# Mapping Digital Storytelling in Interactive Learning Environments

**Ching-Yi Chang** [1,2] and **Hui-Chun Chu** [3,*]

1 School of Nursing, College of Nursing, Taipei Medical University, Taipei 110, Taiwan
2 Department of Nursing, Shuang Ho Hospital, Taipei Medical University, New Taipei 235, Taiwan
3 Department of Computer Science and Information Management, Soochow University, Taipei 100, Taiwan
* Correspondence: carolhcchu@gmail.com; Tel.: +886-915-398218

**Abstract:** This study used bibliographic analysis to validate the research trends of technology-enhanced digital storytelling (DS) in education and educational research published from 2005 to 2022, reviewing the literature found in the Web of Science (WoS) databases. The study investigated trends in learning domains in DS by primarily analyzing the top published journals and the most frequently cited DS-related articles, countries, publishing authors, occurring keywords, research methods, and application domains in the field. This paper mainly used bibliometric analysis and examined the relationship between keywords using the VOSviewer software. Four thematic clusters were identified: DS, education, technology, and literacy. The results show that Robin's study is the most frequently cited DS-related work in the education and educational research field. Important findings in the literature about the main application domains in DS research are that activity is dominant, there is no specific domain, followed by the application of language, and next by position studies. The results highlight the research trend of technology-enhanced storytelling education, and recommendations are provided for future research. For field researchers, the analysis serves as a map to explore the indiscernible relationships among DS research trends, providing new avenues for DS researchers to investigate further.

**Keywords:** digital learning; digital storytelling; literacy; storytelling; research trends; technology

## 1. Introduction

Effective technology has a significant influence on education [1]. Scholars have facilitated researchers and instructors in managing online learning, teaching, and administration on the internet, including how they interact with learners [2]. The development of science and information technology has greatly affected our teaching and interactive methods [3], and the technology used to improve the quality of education is constantly evolving [4]. Many researchers examining COVID-19 are attempting to adopt technology-based blended learning methods to adapt to learners' needs and to improve the global education market [5,6]. The education and educational research field was identified using the Web of Science (WoS) database [7]. In academia, international academic journals on teaching and technology applications related to education and educational research first appeared in 1985 [8]. Since then, technological research for educational purposes has become an increasingly active field and has received increasing attention from scholars. Several recent retrospectives on the use of technology in education describe the current state and development of education and educational research. For example, Rasheed, Kamsin, and Abdullah [9] reviewed distance-related research on online and blended learning. Wood and Shirazi [10] systematically reviewed audience response systems for teaching and learning in higher education to explore the development of higher education over the past few years.

In previous studies, retrospective education and educational research articles were limited to a specific educational domain (such as a discipline or a specific technology) and

were not analyzed using quantitative methods [9,10]. Furthermore, these retrospective studies have limitations, as most used hand-coding methods and involved a tedious and labor-intensive coding process [11]. Therefore, it is necessary to adopt bibliometric methods that are applicable to the acquisition of education and educational research to address the limitations of existing retrospective research and to provide trends and directions in the field of education. Bibliometric analysis is increasingly recognized as a valuable and effective technique for assessing scholarly achievement in a particular field of study [12]. It can be used to understand the content of past research and to predict research trends in specific fields. It has been widely used in scientific research trend analysis and in research on emerging topics within a specific research field [13]. For example, Shen and Ho [14], discussing technology-enhanced learning in higher education, published academic literature on the latent semantic approach. Pinto et al. [15] conducted a quantitative review and illustrated trends in distance education by reviewing scientific production of mobile information publications from 2006 to 2017. Moreover, according to the InCites JCR report, researchers have published a large number of articles in the field of education and educational research on the use of digital technology combined with curricula for innovative teaching and improved learning outcomes. While bibliographic analysis is considered an effective method to illuminate research trends, it can provide researchers or educators with recommendations for future technology-assisted enhanced research developments. Therefore, exploring research trends in the field of education and educational research has a broad research focus and deserves scholars' attention.

Storytelling has been used as a basic form of communication for centuries. Communicating important information, knowledge, and experience to others through stories can help individuals rebuild their lives and experiences, face the learning process, help others to understand the learning process, rebuild their bodies and minds, and give meaning to life [16]. Due to the rise of information technology, students have more opportunities for using different learning strategies to enhance their learning, and the introduction of technology into storytelling has become a teaching strategy and learning tool. Digital storytelling (DS) is a high-level thinking process that can inspire learners to engage in purposeful learning [17]. It is an effective learning design engaging learners in organizing what they have learned during the process of developing digital stories, as indicated by several studies [7,9,10]. The influences of digital storytelling on learners' performances, including critical thinking and other competences, have been reported by several previous studies based on the application results in diverse subjects [18–20]. By learning or editing DS, learners can understand situational events, have the chance to experience the event, learn how to solve problems, organize or construct new knowledge, benefit from the power of healing, and improve their personal resilience and personal awareness. Nair and Yunus [21] conducted a systematic review of 45 articles, and the findings showed that DS is a useful tool to improve students' speaking skills. Walters, Green, Goldsby, and Parker [22] applied DS to teach students problem-solving skills. By combining curriculum design with DS, students have more opportunities to acquire knowledge or experience through the multimodal approach of DS. However, the explosion of information may also affect students' learning. In the era of globalization, DS has been proven to train learners to learn [23]. According to Calik and Seckin-Kapucu [24], combining DS with teaching activities could help students improve their learning effectiveness, face challenges, and actively solve problems.

Based on these previous studies, little research has examined the trend of technology-enhanced DS. If we can understand technology-enhanced DS learning issues and trends, we can gain insight into cross-country research fields, the research interests of core groups, and the evolution of keywords. Thus, this study analyzed the publication trends of DS in SSCI/SCI journals in the education and educational research field from 2020 to 2022. A bibliographic analysis was used to survey the study setting. This study aimed to answer the following research questions:

(1)  What are the top published journals on DS in the education and educational research field?
(2)  What are the most frequently cited DS-related articles in the education and educational research field?
(3)  What are the publishing countries for DS studies in the education and educational research field?
(4)  Which publishing authors have conducted DS studies in the education and educational research field?
(5)  Which keywords frequently occur in research on DS in the education and educational research field?
(6)  What research methods and application domains have been applied in DS studies in the education and educational research field?

## 2. Materials and Methods

### 2.1. Search Criteria

This study examines research articles on technology that apply DS in the education and educational research field that were obtained from Web of Science (WoS) databases and published between 2005 and 2022. The keywords (digital storytelling*) and (technology or e-learning or digital or computer or internet) and (education or learning) were entered into the topic field (1 January 2022). Figure 1 shows the data collection procedure flow. Studies not issued in SSCI journals or not issued during the identified period of 2005–2022 were excluded. Ultimately, 146 articles were retrieved.

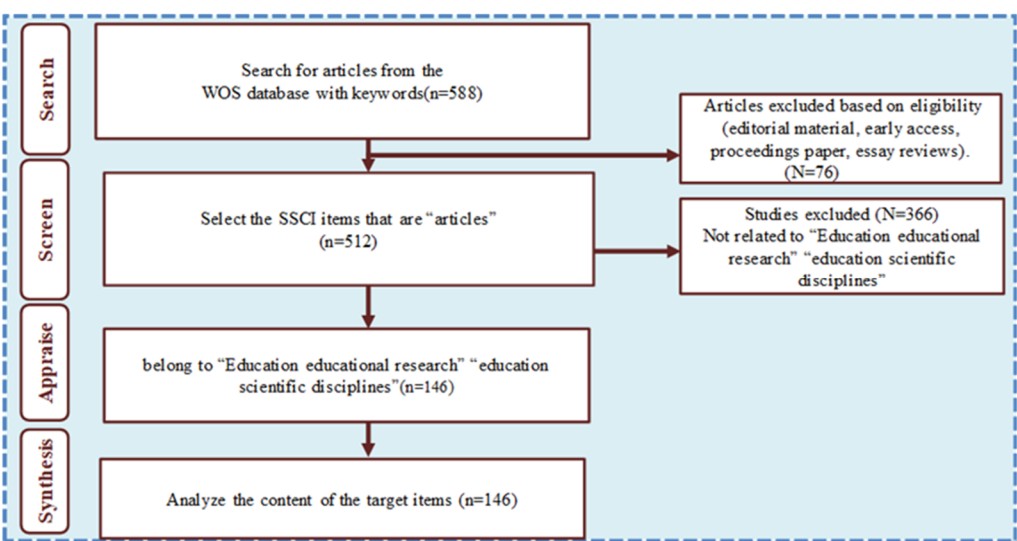

**Figure 1.** Data collection flow.

### 2.2. Data Extraction and Analysis

The bibliometric data of 146 publications were taken from the WoS database. For the data and various citation data collected by the system, the bibliometric analysis of the literature was carried out using an Excel file, and image analysis was carried out using VOSviewer software (v1.6, Leiden University, Leiden, The Netherlands). Through graphical visual analysis, readers can quickly identify the relationship and focus of numbers and texts in data [25], such as keyword co-occurrence analysis. In addition, citation analysis can identify pioneer authors and research in a particular field of knowledge [26]. If the article is frequently cited, it can be reasonably inferred that these authors and research articles are important inspirations for development in this particular field of knowledge and are worthy of follow-up by later scholars [27]. In addition, through the citation rate of articles, scholars can see the historical background of the development of a specific

field of knowledge and inspire academic researchers to find potential research topics and contribution value [28,29].

The frequently occurring keywords of each country's cluster and the authors' cluster were also obtained. To identify DS milestones in education and educational research, issues, learning settings, and educational goals were examined.

## 3. Results

### 3.1. Data Distribution

Figure 2 illustrates the publication conditions of DS articles from 2005 to 2022. From 2005 to 2013, fewer than 10 articles of this type were published each year. This figure rose sharply in 2016. The first study was presented by Ohler [30] for learning in the digital age with DS and was published in the *Journal of Educational Leadership*. As can be seen in Figure 2, due to the advancement of information technology and the development of global mobile devices, schools, and research institutions opened up to the development of DS in digital education in 2016 [31]. At the same time, more and more information technology has been incorporated into the digital curriculum, and the cross-disciplinary educational activities of formal and non-formal education have been introduced.

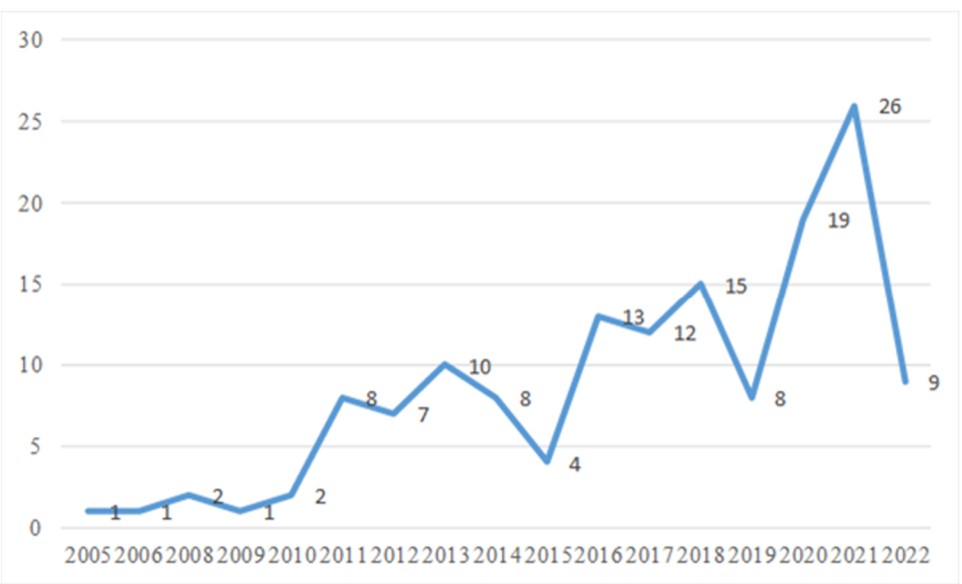

**Figure 2.** No. of articles published per year from 2005 to 2022.

### 3.2. Distribution by Journal

Figure 3 lists the SSCI/SCI journals that have published more than two articles on the topic of digital storytelling between 2005 and 2022. As can be seen in Figure 3, *Computer Assisted Language Learning* and *Learning Media and Technology* published the most articles (seven articles each), followed by *Educational Technology & Society* and the *Journal of Adolescent & Adult Literacy* (six articles each). The outcomes indicate that mainstream DS articles distributed from 2005 to 2022 focused primarily on adopting media to calculate learners' learning consequences. For instance, Svendsen et al. [32] employed DS in education as a learning activity that could foster the migration of critical thinking. Moreover, based on the InCites JCR report, influential journals in the category of education and educational research, *Computer Assisted Language Learning* is a Q1 journal impact factor (IF), ranking 23/265 (Education & Educational Research) in 2022. Among the top rankings in education and educational research, *Computer Assisted Language Learning* is the only educational journal the scope of which includes language learning and technology applications and which has the potential to influence academic development.

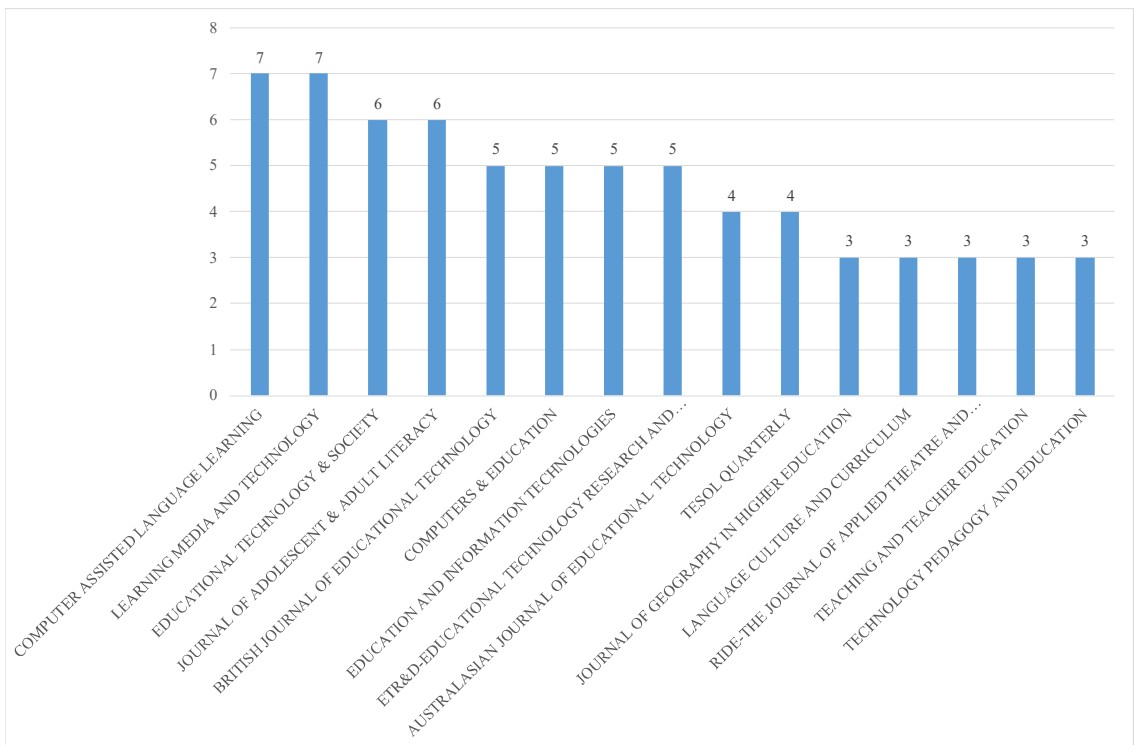

**Figure 3.** Journals with DS article data distribution from 2005 to 2022.

### 3.3. Top Highly Cited DS-Related Articles

Table 1 illustrates the most cited DS-related articles between 2005 and 2022. The most cited article was "A powerful technology tool for the 21st-century classroom" by Robin [33]. The article was cited 390 times and was published in *Theory into Practice*. The next most cited article was "A meaningful technology-integrated approach for engaged student learning" (279 citations) by Sadik [34], which was published in *ETR&D-Educational Technology Research and Development*. The third most cited article was "Digital storytelling for enhancing student academic achievement, critical thinking, and learning motivation" (223 citations) by Yang and Wu [35], which was published in *Computers & Education*. These highly cited DS-related articles could explain the powerful impact of DS on learning in the education and educational research area.

**Table 1.** List of the top 10 highly cited DS-related articles from 2005 to 2022.

| Publication Source | Authors | Title | Number of Citations |
|---|---|---|---|
| Theory into Practice | Robin [33] | A powerful technology tool for the 21st-century classroom | 390 |
| ETR&D-Educational Technology Research and Development | Sadik [34] | A meaningful technology-integrated approach for engaged student learning | 279 |
| Computers & Education | Yang & Wu [35] | Digital storytelling for enhancing student academic achievement, critical thinking, and learning motivation | 223 |
| Research in the Teaching of English | Hull & Katz [36] | Crafting an agentive self: Case studies of digital storytelling | 191 |
| Language Learning & Technology | Hafner & Miller [37] | Fostering learner autonomy in English for science: a collaborative digital video project in a technological learning environment | 132 |

**Table 1.** *Cont.*

| Publication Source | Authors | Title | Number of Citations |
|---|---|---|---|
| Educational Technology & Society | Hung, Hwang, & Huang [38] | A project-based digital storytelling approach for improving students' learning motivation, problem-solving competence, and learning achievement | 118 |
| Educational Leadership | Ohler [30] | The world of digital storytelling | 96 |
| Computers & Education | Phan, McNeil, & Robin [31] | Students' patterns of engagement and course performance in a Massive Open Online Course | 81 |
| Educational Technology & Society | Xu, Y.,Park, & Baek [39] | A new approach toward digital storytelling: an activity focused on writing self-efficacy in a virtual learning environment | 77 |
| Health Education Research | Blue Bird Jernigan et al. [40] | Addressing food insecurity in a Native American reservation using community-based participatory research | 74 |

### *3.4. Country's Data Distribution*

Figure 4 shows which countries have published the most DS articles based on the nationality of the first author. The outcomes demonstrate that researchers from various countries have applied DS in education and educational research. Figure 4 shows the top countries/regions. The United States has published the most DS articles (42), followed by Taiwan (18 articles), Australia (15 articles), Canada (11 articles), and Turkey (10 articles).

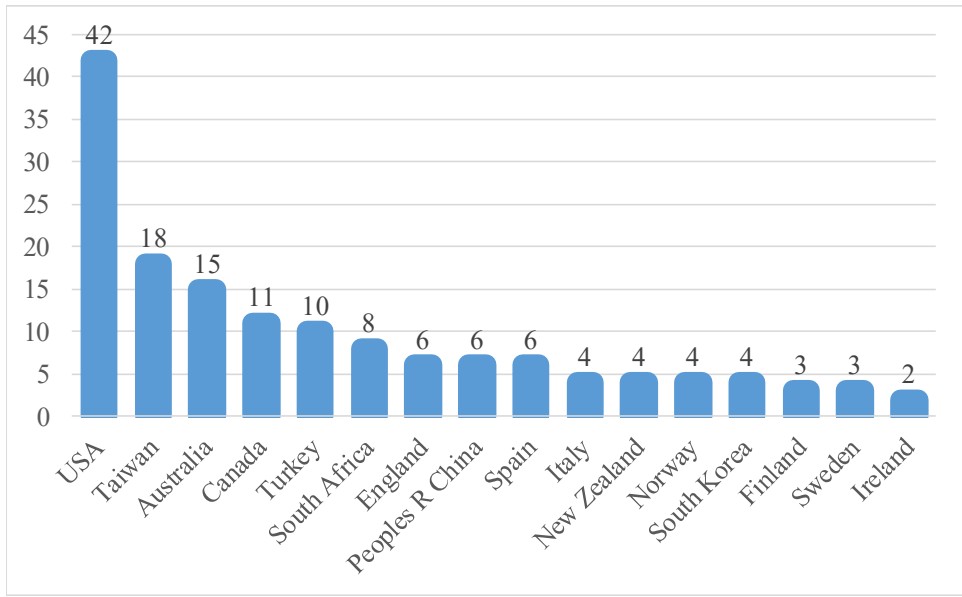

**Figure 4.** DS-related articles published from 2005 to 2022 by country.

### *3.5. Author Efficiency*

Figure 5 presents the authors who published the most DS articles from 2005 to 2022. The top five authors are Liu, C. C., (five articles) from Taiwan; Gachago, D., from South Africa (four articles); Condy, J., from South Africa; Ivala, E., from South Africa; and Robin, B. R., from the USA (with three articles each).

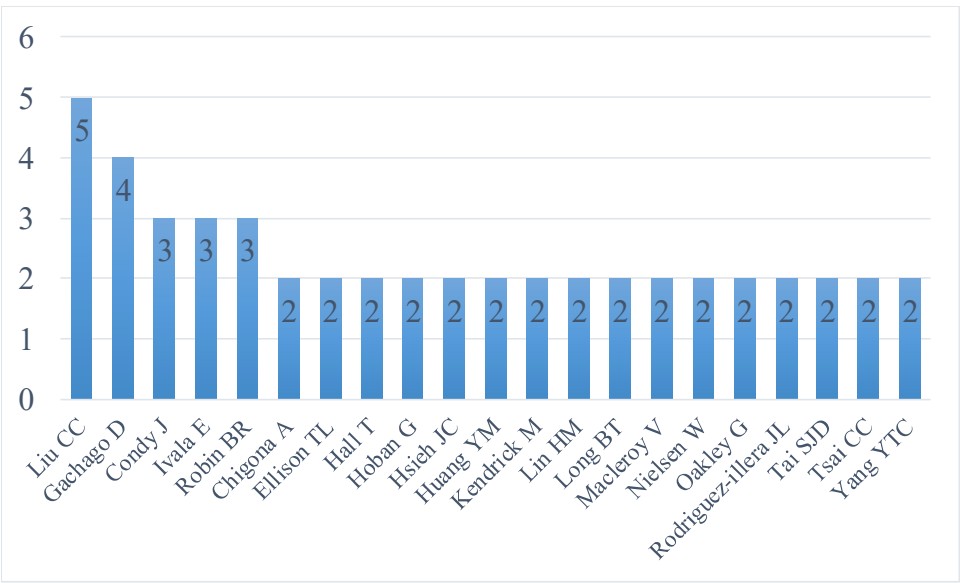

**Figure 5.** Authors who published the most DS articles between 2005 and 2022.

*3.6. Research Method*

Figure 6 shows the research methods implemented in the DS studies. The majority of the studies utilized qualitative methods (77 articles); followed by quantitative methods (34 articles); position studies, such as theoretical or descriptive analysis (23 articles); and mixed methods (12 articles).

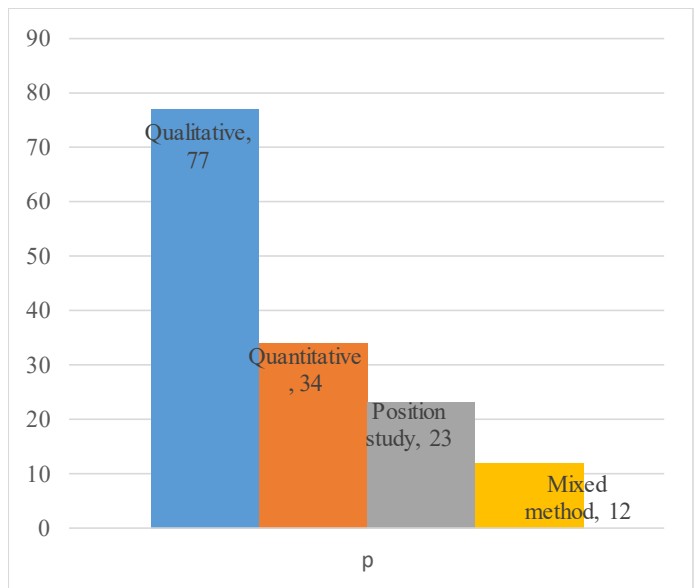

**Figure 6.** DS-related articles published between 2005 and 2022 by research method.

Figure 7 shows the design of the DS-related empirical studies, which increased considerably between 2019 and 2021 when a mainstream DS study used a qualitative method to analyze the impact of DS on learners' engagement with a class activity or object design. For instance, Hausknecht, Freeman, Martin, Nash, and Skinner [41] designed an intergenerational DS workshop using indigenous knowledge to bring elders and schoolchildren together to co-create DS. The study also used a community-based participatory research approach via interview methods.

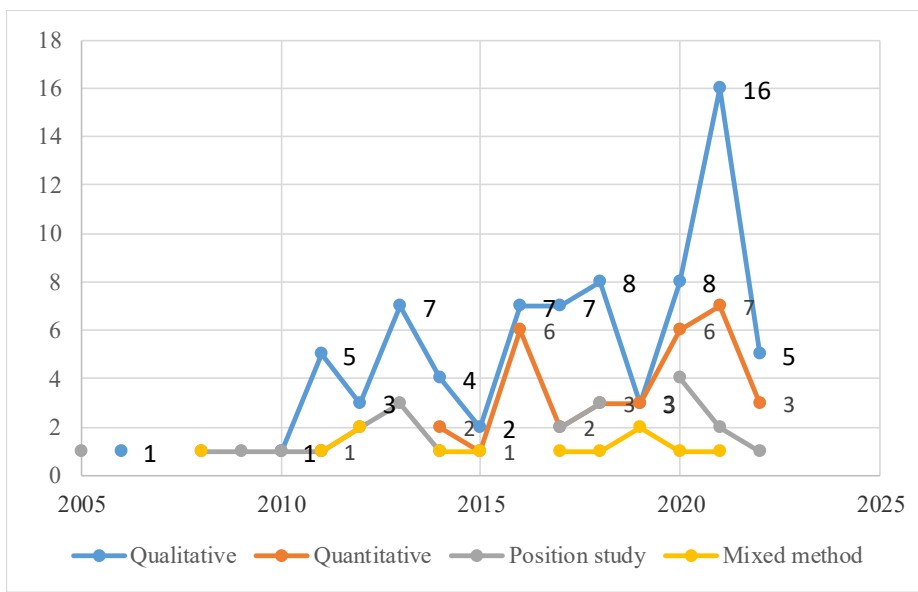

**Figure 7.** DS-related article design from 2005 to 2022.

### 3.7. Application Domain

Figure 8 presents the results of an analysis of the application of DS. From 2005 to 2022, excluding other DS activity/courses in the education and educational research field (49 articles), DS was most frequently applied to "language" activities/courses (47 articles), followed by "position studies" (21 articles, e.g., non-specified, DS concept or opinion), "science" (13 articles), and "preservice teacher courses" (8 articles). The results of the analysis also revealed that DS was applied to a wide range of applications, including ecology, geography, chemistry, health courses, psychology, customer services in the workplace, and nursing education, hinting at the potential impact of future research and its application.

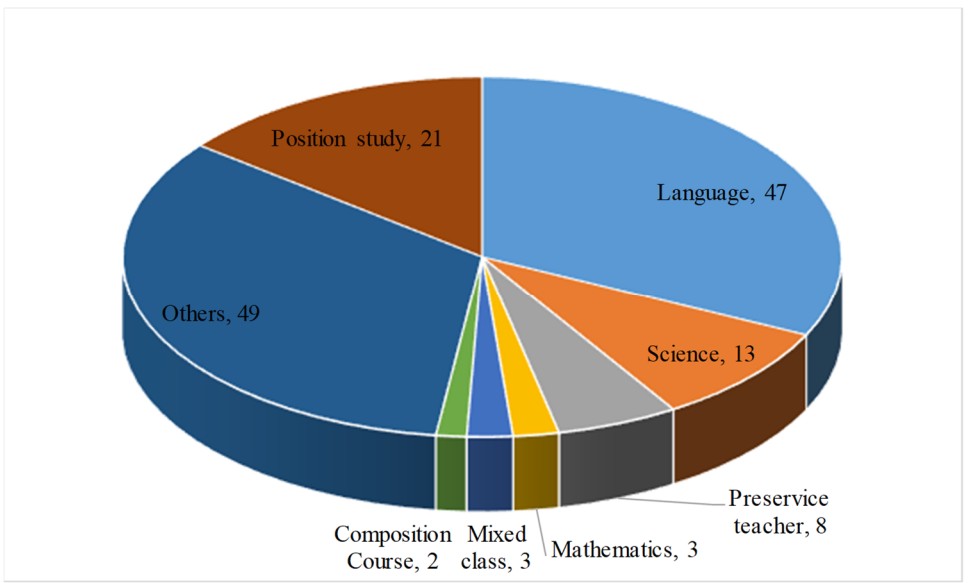

**Figure 8.** Distribution of DS to course type from 2005 to 2022.

### 3.8. Popular Keywords

To explore global trends and examine the keywords most commonly used by scholars, co-occurrence citation analysis and author keywords were selected. Using the VOSviewer software system, the minimum number of occurrences of a keyword was set to five, and the number of selected keywords was automatically expressed as 71. Figure 9 presents the

results of the dissemination of the co-occurrence keywords. The most popular keywords based on the number of occurrences were DS (78), education (17), technology (16), and literacy (13). This finding indicates that most inquiry productions are associated with the presentation of DS in education and technology learning environments.

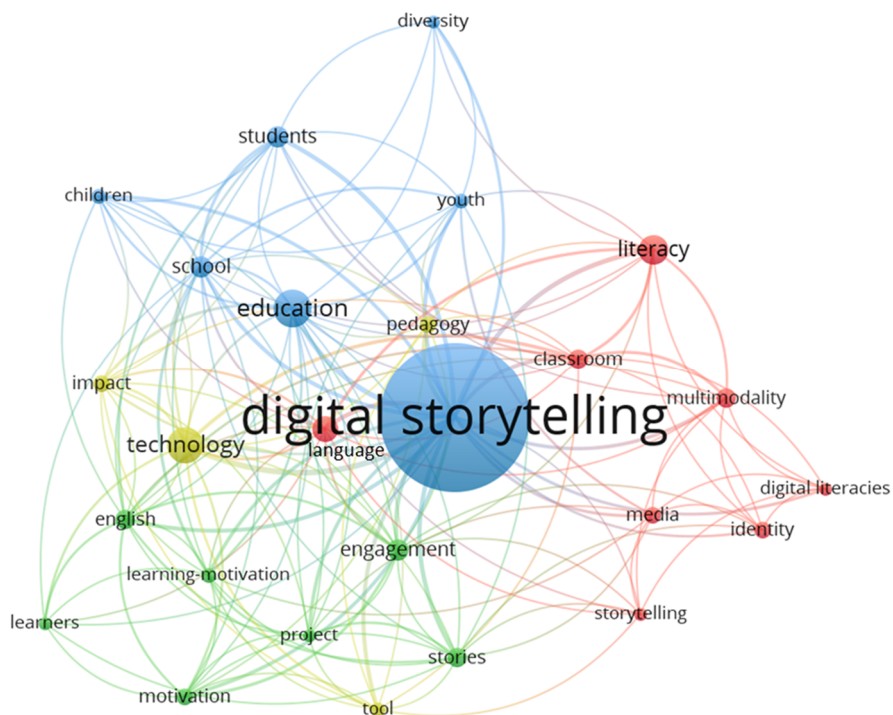

**Figure 9.** Dissemination of the most popular keywords.

Figure 9 shows that the most-used keyword map comprises four clusters. Table 2 displays the commonly used keywords in each cluster. The most solid keyword in the first cluster (red cluster) was storytelling. The first (dark blue) cluster typically discussed the use of storytelling in learning environments in diverse educational settings. The second (yellow) cluster's strongest keywords are technology, impact, pedagogy, and tool. Associated with other clusters, the research in this cluster focused more on the influence of technology on students' performance. Furthermore, the third (red) cluster's research mostly discussed literacy, classroom, digital literacies, identity, language, media, multimodality, and storytelling. The fourth (green) cluster discussed learning engagement, English, learners, learning motivation, motivation, projects, and stories in the DS learning environment. Specifically, this cluster used technology to record learning engagement and further investigate learning motivation in DS learning environments. These clusters could represent future research trends.

**Table 2.** The keyword clusters through co-occurrence analysis (sorted in occurrence order).

| Cluster 1 | Cluster 2 | Cluster 3 | Cluster 4 |
| --- | --- | --- | --- |
| digital storytelling | technology | literacy | engagement |
| education | tool | language | English |
| diversity | impact | classroom | learners |
| children | pedagogy | digital literacies | learning-motivation |
| school | | identity | motivation |
| students | | media | project |
| youth | | multimodality | stories |
| | | storytelling | |

## 4. Discussion

In recent years, due to the popularization of information technology, learners' learning methods and resources have become more diverse. In particular, the use of DS design and mobile devices for teachers' courses has provided innovative teaching opportunities [42]. This study analyzed the most cited SSCI/SCI studies on DS in the WoS database to identify essential articles, journals, authors, countries, keywords, and domains. We performed a quantitative analysis to identify the associations between DS study design and novel education activities. In a generation characterized by emerging global technology, teachers are using DS in teaching activities to help students organize and follow knowledge research trends in education.

As shown in Figure 2, the overall trend of DS in education and educational research has grown significantly since 2019, as technology has progressed. Proposing DS-related investigations will impact the global education era. Among the journals publishing articles on DS, we found that *Computer Assisted Language Learning and Learning Media* and *Technology* were the top journals in education and educational research, followed by *Educational Technology & Society* and the *Journal of Adolescent & Adult Literacy*. These four journals also published the most cited articles, and the five-year IFs of these SSCI/SCI journals were 4.789, 4.621, 3.941, and 1.741, respectively. According to Kantek and Yesilbas [43], these journals are well thought out and often cited for their high-quality research. This assessment aligns with our results. These findings suggest that researchers are willing to invest in these core journals to maximize their impact and present their DS research results [32,44]. Scholars, such as Jimenez et al. [44], have applied DS to education by negotiating youth–adult relations and improving their digital media literacy, and have published their articles in *Learning, Media and Technology*, which is in the education and educational research field.

In terms of the most frequently cited DS-related articles, we found that the highest citation rate occurred in 2008 (390 citations) for Robin's study [33]. He proposed a powerful technology tool related to DS for the 21st-century classroom. According to Robin's study, DS could be applied as a teaching instrument to improve participant learning results and impact the education and educational research field. The next most cited study (279 citations) was by Sadik [33], who published "Helping teachers develop teaching through digital technology DS", which applied DS to teaching tasks to help students learn. Hava [45], Mertala [46], and Kim and Hall [47] cited Sadik's [34] research and proposed a study design proving that DS is an effective strategy for improving student learning outcomes. Sadik's [34] comprehensive DS quality design may be a factor in its high number of citations. This finding is consistent with the findings of Su and Chang [48]. As shown in Table 1, the top 10 most cited studies had between 74 and 390 citations. Researchers could track studies with high citations and incorporate their work into their own research.

Regarding the publishing country and authors' analysis, consistent with Ivanović and Ho's [49] research findings, the United States is the most-published country in the education and educational research field based on the nationality of the first author. As Figure 5 shows, the most-published author, Liu, C. C., produced five papers in 2014, 2016, 2017, 2018, and 2019, and conducted a series of studies related to DS in English classes to boost students' learning performance. Liu, Tai, and Liu [50] applied DS to enhance English classes and facilitate students' learning motivation and performance. Liu, Yang, and Chao [51] used a longitudinal study to analyze learners' participation in collaborative DS activities. They proposed that DS has strong potential to support novel teaching and improve learners' performance.

As for the research methods, the study's findings were consistent with Su and Chang [48] in its focus on most research methods, including qualitative and quantitative methods. For example, in terms of qualitative methods, Büyükkarci and Müldür [52] recruited primary school teacher candidates in mathematics and used a qualitative research program for content analysis. They stated that primary school teacher candidates should pay attention to preparatory DS work while creating digital stories in class. In terms of quantitative methods,

Molan, Weber, and Kor [53] applied DS to an immersive virtual learning environment and used standard descriptive statistics, a chi-square test of independence, paired-samples *t* tests, independent samples *t* tests, and McNemar's tests to analyze children's knowledge, as well as to show that DS could improve learning outcomes. Other methods of design include qualitative and quantitative mixed methods. Olitsky et al. [54] compared the DS of college and high school students in authentic science and found increased communication among students. Thus, future research should consider how to determine a quasi-experimental design or randomized trials associated with DS innovation design, which may provide new insights into these fields. In addition, the application domains, as shown in Figure 8, and other DS activities/courses in the education and educational research field are the most common domains. For example, Liu, Tai, and Liu [50] used DS in EFL class activities and efficiency to improve students' learning motivation and performance. We could say that DS is rarely applied in various fields, such as in specific education domains. Thus, there was a chance to progress and implement DS scenarios during the participants' learning advancement.

Regarding frequently occurring keywords, such keywords assist researchers in finding related studies in the database. From the results of this study's analysis, we found that the main authors' keywords focused on four main areas: DS, education, technology, and literacy. For example, in terms of "digital storytelling" with "education," Çetin's [17] work boosted preservice teachers' digital literacy. Chen Hsieh and Lee [55] used DS in EFL education to evaluate student motivation and satisfaction and to improve English learning performance. Accordingly, a keyword inquiry could contribute to the research trend. Meanwhile, the keywords that are most commonly used indicate the most interesting topics for researchers [29]. For instance, Churchill [42] used DS with the mobile device method, which allows students to conduct research, collect and analyze data, and present their DS findings, as well as raise students' digital literacy skills. Furthermore, Çetin [17] illustrated DS in teacher education and its impact on preservice teachers' digital literacy. This study obtained the most-cited research on DS in education and educational research. Thus, the results of these studies may inform follow-up research to understand the global emphasis on prioritizing DS research in education.

In addition, according to the analysis results of this paper, DS is a powerful tool and teaching strategy. Through the editing of DS situations or the transmission of DS teaching situations, participants can construct knowledge and organize knowledge to increase their learning. In addition, the journal analysis confirms that *Computer Assisted Language Learning* is not only a well-known journal with a recognized academic impact on education and educational research but also contributes to the transformation of technology for in-person and online education. Interactively, Zhang et al. [56] explored trends related to online learning, teaching, and administration on the internet. For the above reasons, teaching theories, teaching outcomes, and applied perspectives on educational practice can help identify research issues in innovative education. The survey results indicate that research on DS is worth continuing. With the advancement of science and technology, future studies could investigate how educators can combine a variety of innovative resources to teach learners to learn. From the keywords analyzed by VOSviewer, it was also found that the use of DS as a learning strategy or tool in the classroom environment can heighten learners' motivation and engagement, especially the cultivation of learners' information literacy by DS, which is also a response to global informatization. Even though information and learning environments change over time, educational theories remain the same; thus, researchers should discuss DS strategies in depth, such as through DS editing and workshop management. Finally, this study encourages future education researchers to design content analyses that include quantitative or qualitative content to discuss international mobile trends and implications for DS research from multiple perspectives.

*Strengths and Limitations*

This research's main contribution was a bibliometric analysis of DS research in the education and educational research field using VOSviewer to examine popular keywords.

This study is the first bibliometric analysis to define DS research in the education and educational research field. Nonetheless, there are some limitations that should be noted. First, the study was limited to the results obtained from a single database. Using a similar method with different databases may produce diverse outcomes. Second, the study found many DS articles distributed between 2005 and 2022, with an increasing trend in 2019. In addition, the study only used author keywords, which may limit the focus to the education and educational research field. Searching all fields in the database would likely present new DS trends. Third, the most frequently cited articles in this research addressed DS in the education and educational research field. However, DS has increasingly been applied in various disciplines.

## 5. Conclusions

From the research work, the empirical benefits of DS apply to a variety of different issues, and the topics covered are reported in this study. Instructors should discuss with participants, identify their learning outcomes, have an opportunity to debrief, and address the skills and knowledge gained. Participants can learn from creating their own DS content, experiencing stories from the perspective of digital creators and viewers alike, listening to digital stories from peers, and from the DS shown to other members. Learning by co-creating a digital story between the activities and design can facilitate the overall effect. Further studies should focus on the specific changes in learners' behavior caused by implementing DS with technology and literacy in multidisciplinary training.

**Author Contributions:** Conceptualization, C.-Y.C. and H.-C.C.; methodology, H.-C.C.; software, validation, and formal analysis, C.-Y.C. and H.-C.C.; investigation, C.-Y.C. and H.-C.C.; resources, C.-Y.C. and H.-C.C.; data curation, C.-Y.C. and H.-C.C.; writing—original draft preparation, C.-Y.C. and H.-C.C.; writing—review and editing, C.-Y.C. and H.-C.C.; visualization, C.-Y.C. and H.-C.C.; supervision, H.-C.C.; project administration, H.-C.C. All authors have read and agreed to the published version of the manuscript.

**Funding:** This research was funded by the Ministry of Science and Technology of Taiwan, grant number MOST 111-2410-H-038-029-MY2 and MOST-109-2511-H-011-002-MY3.

**Institutional Review Board Statement:** Not applicable.

**Informed Consent Statement:** Not applicable.

**Data Availability Statement:** Not applicable.

**Conflicts of Interest:** The authors declare no conflict of interest.

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
