# Peer review of "Mapping Digital Storytelling in Interactive Learning Environments"

_sustainability, doi:10.3390/su141811499_

Round 1

Reviewer 1 Report

The article respect the methodology of writing a scientifical article and I congratulate the author in this regard.

In my opinion "Digital Storytelling" is not an appropriate subject for an scientifical article, because by itself do not solve any problems. 

I would advise the authors to reshape the research trying to find out, for example:

- how Digital Storytelling might influence critical thinking,  

- does Digital Storytelling have any influence of transversal skills? are there any subfactors that have great impact on transversal skills? Which are they?

- how Digital Storytelling enhance innovation in research project and their implementation

- which is the positive economical/ informational impact of Digital Storytelling

and so on.

Another weakness: the authors cite old sources, but talks about COVID context.... 

Author Response

We appreciate the valuable comments from the reviewers and have revised the paper according to the suggestions. Please refer to the attached revision summary.

Thank you.

Reviewer 2 Report

The authors report on a bibliographic analysis regarding digital storytelling, a teaching methodology of proven values to students’ learning and engagement. Given the advancement of technology and a growing amount of research studies, periodic review of digital storytelling is essential. The current work involving the bibliographic analysis and recommendations derived would be of interest to identify the contemporary research trend and important issues for future research. In general, the study is a well written one with good support of references. There are clear indications of research questions and results are also further presented with a variety of figures and tables. For further consideration, there are some issues as suggested below.

1.

Ln 83-84: the reference by Nair and Yunus [19] is appropriate, but I think the description there should be changed to be more specific with regards to the review nature of the cited study. The original statement “Nair and Yunus [19] applied DS to teach communication skills to students” is suggested to be replaced as something like: “Nair and Yunus [19] conducted a systematic review of 45 articles and the findings showed that DS is a useful tool to improve students’ speaking skills”.

2.

Ln 297: suggest to add the information “based on the nationality of the first author” after “the US is the most-published country”.

Ln 322-324: can extend the discussion here for mentioning the need to extend the applications and it’s worthwhile for studies addressing wider disciplines or different specific subjects, as also described with reference to ln 216-220.

3.

According to ln 252 and table 2 on page 10, “language” is one of the identified items in the co-occurrence analysis, but the corresponding label is missing in the mapping as shown in Figure 9 (page 9).

4.

Ln 269-270, “As shown in Figure 1…”, the descriptions are taken reference from Figure 2 instead of Figure 1.

5.

I would like to seek clarification or updates from authors regarding Figure 4 on page 7 which shows the countries based on the nationality of the first author. The numbers for each country add up to 157, which is larger than the number of studies analyzed (146 according to ln 119 and Figure 1 on page 3.

6.

Page 5, Table 1: the name of the second column: “Authors/year”, should the word “year” be deleted or should the publication year be added in each item? Currently, the whole column does not indicate any year.

In the same table, regarding the third column, would it be better to be named as “Title” instead of “Research Foci”, as the descriptions are the names of the articles?

7.

Ln 179: The name of Table 1 is wrong. It should not be “The keywords cluster through co-occurrence analysis”, which is irrelevant with the information shown in table 1 regarding the details of the top-cited articles.

8.

The captions of the figures should be refined. For example, some of the captions are similar  (“Figure 2. Distributed articles on DS from 2005 to 2022” and “Figure 3. DS article data distribution from 2005 to 2022”), suggested to be more specific to the content displayed. For example, ln 150 regarding Figure 2’s caption, would authors consider specifying as “No. of articles published per year from 2005 to 2022”.

9.

Further editing work is suggested. For example,

Ln 25: literacy is also suggested to be included as it is one of the identified clusters and which have also been discussed in the manuscript.

Ln 58, a close bracket “ ] ” is missing.  

Ln 115: please check if the year 1978 and 2022 is correct.

Ln 153: the conjunction word “and”, between the two journal names (which are in italics), should not be made italics. Similarly for ln 272.

Ln 155-156: to be consistent with the names displayed in Figure 3 and in the official websites, the names of the journals would be written as “Educational Technology & Society”, and “Journal of Adolescent & Adult Literacy”. Similarly for ln 273 and 274.

Page 9, Figure 8, there are overlapping of words and numbers in the figure regarding preservice teacher and mathematics.

Ln 235-244 is duplicated with ln 223-232, please remove either one. 

Author Response

(The authors gave the same response as above.)

Round 2

Reviewer 1 Report

The authors answered to some of my concerns.